# The Arabic Version of the Impact of Event Scale-Revised: Psychometric Evaluation among Psychiatric Patients and the General Public within the Context of COVID-19 Outbreak and Quarantine as Collective Traumatic Events

**DOI:** 10.3390/jpm12050681

**Published:** 2022-04-24

**Authors:** Amira Mohammed Ali, Rasmieh Al-Amer, Hiroshi Kunugi, Elena Stănculescu, Samah M. Taha, Mohammad Yousef Saleh, Abdulmajeed A. Alkhamees, Amin Omar Hendawy

**Affiliations:** 1Department of Psychiatric Nursing and Mental Health, Faculty of Nursing, Alexandria University, Smouha, Alexandria 21527, Egypt; naeem.saleem2@gmail.com; 2Faculty of Nursing, Isra University, Amman 11953, Jordan; r.al-amer@outlook.com; 3School of Nursing and Midwifery, Western Sydney University, Penrith, NSW 2751, Australia; 4Department of Psychiatry, Teikyo University School of Medicine, 2-11-1 Kaga, Itabashi-ku, Tokyo 173605, Japan; hkunugi@med.teikyo-u.ac.jp; 5Department of Mental Disorder Research, National Institute of Neuroscience, National Center of Neurology and Psychiatry, 4-1-1 Ogawahigashi, Kodaira, Tokyo 187-8502, Japan; 6Faculty of Psychology and Educational Sciences, University of Bucharest, 050663 Bucharest, Romania; elena.stanculescu@fpse.unibuc.ro; 7Psychiatric and Mental Health Nursing Department, Faculty of Nursing, Mansoura University, Mansoura 30016, Egypt; dr.samah_hany@yahoo.com; 8Clinical Nursing Department, School of Nursing, The University of Jordan, Amman 11942, Jordan; m.saleh@ju.edu.jo; 9Department of Medicine, Unayzah College of Medicine and Medical Sciences, Qassim University, Unayzah 52571, Al Qassim, Saudi Arabia; 10Department of Animal and Poultry Production, Faculty of Agriculture, Damanhour University, Damanhour 22516, Egypt; amin.hendawy@gmail.com

**Keywords:** Coronavirus Disease-19, COVID-19, the Impact of Event Scale-Revised (IES-R), post-traumatic stress disorder, psychiatric patients, the general public, healthy individuals, quarantine, gender differences, confirmatory factor analysis, measurement invariance, differential item functioning, psychometric evaluation, concurrent validity, convergent validity, discriminant validity/known-group validity, Arabic/Saudi Arabia

## Abstract

The Coronavirus Disease-19 (COVID-19) pandemic has provoked the development of negative emotions in almost all societies since it first broke out in late 2019. The Impact of Event Scale-Revised (IES-R) is widely used to capture emotions, thoughts, and behaviors evoked by traumatic events, including COVID-19 as a collective and persistent traumatic event. However, there is less agreement on the structure of the IES-R, signifying a need for further investigation. This study aimed to evaluate the psychometric properties of the Arabic version of the IES-R among individuals in Saudi quarantine settings, psychiatric patients, and the general public during the COVID-19 outbreak. Exploratory factor analysis revealed that the items of the IES-R present five factors with eigenvalues > 1. Examination of several competing models through confirmatory factor analysis resulted in a best fit for a six-factor structure, which comprises avoidance, intrusion, numbing, hyperarousal, sleep problems, and irritability/dysphoria. Multigroup analysis supported the configural, metric, and scalar invariance of this model across groups of gender, age, and marital status. The IES-R significantly correlated with the Depression Anxiety Stress Scale-8, perceived health status, and perceived vulnerability to COVID-19, denoting good criterion validity. HTMT ratios of all the subscales were below 0.85, denoting good discriminant validity. The values of coefficient alpha in the three samples ranged between 0.90 and 0.93. In path analysis, correlated intrusion and hyperarousal had direct positive effects on avoidance, numbing, sleep, and irritability. Numbing and irritability mediated the indirect effects of intrusion and hyperarousal on sleep and avoidance. This result signifies that cognitive activation is the main factor driving the dynamics underlying the behavioral, emotional, and sleep symptoms of collective COVID-19 trauma. The findings support the robust validity of the Arabic IES-R, indicating it as a sound measure that can be applied to a wide range of traumatic experiences.

## 1. Introduction

Coronavirus Disease-19 (COVID-19) is associated with high infectivity and fatality, creating a collective crisis in the global community [1,2]. Apart from loneliness and distress resulting from COVID-19 lockdowns [2,3], terrorizing images of the outbreak and fake information communicated by mass media heighten psychological distress, intensify traumatic emotions, and accelerate fear levels regarding the negative effects of the pandemic on health and everyday life, with increased universal occurrence of COVID-19-related post-traumatic stress disorder (PTSD) [2,3,4,5,6,7]. PTSD is associated with increased mental symptomatology (e.g., anxiety and depression) and maladaptive behaviors (e.g., illicit drug use and suicidality) [8,9]. Such adverse effects have been reported during the COVID-19 outbreak among youth and older adults, especially those with chronic physical and mental disorders [10,11,12]. Emotional resolution of traumatic experiences may be complicated by associated and/or pre-existing psychological symptoms [13,14]. Therefore, adequate measurement of PTSD symptoms may allow timely identification and management of individuals vulnerable to trauma and related psychological disability [15].

The Impact of Event Scale-Revised (IES-R) is a common measure of the severity of PTSD key features: intrusion, avoidance, and hyperarousal [8,16,17]. It has been translated into many languages, such as Italian [8], Japanese [18,19], Chinese [20], French [21], Greek [22], Norwegian [17], Spanish [23], and Arabic [24]. Nonetheless, reports on its dimensionality are inconsistent both in English [16,17,25] and translated versions [23,24,26]. A few studies replicated the three-factor structure [9,21,27], while other studies reported best fits of two-factor (avoidance and intrusion/hyperarousal) [16,19], four-factor (intrusion, avoidance, hyper-arousal, and numbing or sleep disturbance) [23,25], and five-factor structures (intrusion, avoidance, hyperarousal, numbing, and sleep disturbance) [25,26]. Models comprising more than three factors were improved through error correlations [25], suggesting that the IES-R may contain more undetected factors [25]. Studies of measurement invariance of the IES-R are scarce, and they report metric and scalar non-invariance across groups (e.g., occasions within the emergency room and gender) [20,25]. This necessitates further evaluation of the invariance of the IES-R to ensure valid comparison of PTSD symptomatology across groups.

The Arabic version of the IES-R has been previously tested via exploratory factor analysis (EFA) in 40 Middle Eastern refugees in Australia [24]. This small sample fails to meet the minimum sample size required to perform a reliable EFA, which casts doubt on the credibility of the findings. Given that the literature is inconsistent regarding the structure [16,23,24,25] and invariance of the IES-R [25], the current study aimed to investigate the factor structure of the Arabic version of the IES-R in relevance to the current COVID-19 crisis in a sample from quarantine settings in Saudi Arabia (quarantine sample), a clinical sample of community-dwelling psychiatric patients (sample 1), and a sample from the general public (sample 2). We also assessed its measurement invariance across gender, age, and marital status. Criterion validity was examined by assessing the association of the IES-R with the Depression Anxiety Stress Scale-8 (DASS-8) and its subscales as measures of psychological distress, depression, anxiety, and stress [28,29]. We hypothesized that the IES-R will positively correlate with the DASS-8 and its subscales. The IES-R was also expected to correlate with perceived vulnerability to COVID-19 and perceived physical health status. Hyperarousal and intrusion are reported to shape the mental and behavioral aspects of COVID-19 trauma [7]. Therefore, we conducted a path analysis model to examine the mechanism through which cognitive activation may affect sleep, mood, and avoidance aspects of COVID-19 trauma in our samples.

## 2. Materials and Methods

### 2.1. Study Design, Participants, and Procedure

This cross-sectional study is a secondary analysis based on three convenience samples collected from Saudi Arabia during the lockdown period (29 April until 19 May 2020). The first sample (N = 214) was collected from seven quarantine settings in Riyadh and Qassim. It comprised travelers returning back to Saudi Arabia, with suspected COVID-19, or mild cases. This sample is referred to herein as the quarantine sample. In Saudi Arabia, quarantine settings are hotels designated for quarantine purposes. The capacity of each setting varies from 40 to 200 rooms. They are equipped with doctors, nurses, social workers, and psychologists. Each individual is placed in a single room alone for a planned period of 14 days. The residents are observed for vital signs at least twice a day, and they receive necessary medication prescribed by the doctor on duty [30].

A sample of 1160 community-dwelling individuals was collected through invitations disseminated via WhatsApp and Twitter groups. Sample 1 comprised 168 respondents who testified having a pre-existing mental disorder diagnosed by a psychiatrist. Sample 2 comprised 992 individuals testifying an absence of mental diseases. From all respondents, including those in quarantine, data were collected through an anonymous online survey delivered through Google Forms. Individuals were included in the study if they were 18 years or older, could speak Arabic, and signed a digital informed consent. Further details are described elsewhere [6,29].

### 2.2. Study Instruments

The online questionnaire consisted of several parts. The first part assessed sociodemographic and clinical characteristics of the participants. It contained questions about gender, age, education, and having a medical diagnosis of a pre-existing mental health problem. Accordingly, participants reporting a diagnosis of a psychiatric disorder were included in the clinical sample (sample 1) while participants reporting an absence of a clinical diagnosis were included in the general public sample (sample 2).

The second part consisted of the IES-R. The IES-R comprises 22 items in three subscales, which describe the major features (intrusion, avoidance, and hyperarousal) of PTSD relevant to a specific trauma—COVID-19 in the current context. An example intrusion item is “thought of it when I didn′t mean to” (item 6). An example avoidance item is “tried not to think about it” (item 11). An example hyperarousal item is “was watchful or on-guard” (item 21) [16]. The internal consistency of the IES-R is good (α = 0.93) with a test–retest reliability of 0.95 [25]. We obtained this Arabic version of the IES-R (Appendix A) from Davey and co-workers who translated the scale into Arabic and back-translated it into English according to standards declared by the World Health Organization. They reported good reliability of the IES-R (α = 0.93) and its subscales: intrusion (α = 0.77), avoidance (α = 0.75), and hyperarousal (α = 0.86) [24].

The third part of the questionnaire consisted of the validated Arabic version of the DASS-21. This scale consists of three subscales, which measure symptoms of depression, anxiety, and stress. Each subscale consists of 7 items. Items are rated on a 4-point scale that ranges from 0 (did not apply to me at all) to 3 (applied to me very much or most of the time). The DASS-21 demonstrates good internal consistency (α = 0.88) [31]. However, it does not discriminate mental symptoms underlying the tripartite model (depression, anxiety, and stress) [31,32,33]. Discriminant validity tests uncovered better discrimination among the subscales of the DASS-8, a shortened version of the DASS-21, than the parent scale [28]. Therefore, we included the DASS-8 in the present analysis for criterion validity testing. The DASS-8 comprises eight items in three subscales: depression (three items, e.g., had nothing to look forward), anxiety (three items, e.g., felt close to panic), and stress (two items, e.g., found it difficult to relax). The minimum score of the DASS-8 and its subscales is 0 while their maximum scores are 24, 9, 9, and 6, respectively [28,29]. The reliability of the DASS-8 and its subscales ranges from excellent to very good in sample 1 (α = 0.94, 0.85, 0.89, 0.84, respectively) and sample 2 (α = 0.91, 0.79, 0.79, 0.80, respectively) [29].

A single question was used to assess perceived health status “rate your physical health status on a scale from 1 = very bad to 5 = very good”. One question was used to assess participants′ perceived vulnerability to COVID-19 “rate your perceived vulnerability to COVID-19 on a scale from 1 = very unvenerable to 5 = very vulnerable”.

### 2.3. Ethical Considerations

The study plan was approved by the Institutional Review Board of Al Qassim University (No.19-08-01). Participants who accessed the questionnaire were first introduced to a consent form confirming that the purpose of data collection was purely scientific and that participation was voluntary as it would help the researchers explore the psychological impact of the COVID-19 outbreak. The form presented information on the average time required to complete the questionnaire. It also noted that the questionnaire was anonymous and emphasized that collected personal data would not be disclosed, shared, or communicated to anyone.

### 2.4. Statistical Analysis

The dimensionality of the Arabic version of the IES-R was tested in the quarantine sample using EPA, with maximum likelihood method of extraction, varimax rotation, Kaiser–Meyer–Olkin (KMO) measure of sampling adequacy, and Bartlett′s test of sphericity. In addition, CFA (maximum likelihood with bootstrapping based on 2000 random replications) was used to examine data fit to a number of competing models in sample 1 and sample 2. Based on previous research that assessed the latent factor structure of the IES-R, we tested the theory-based three-factor structure as well as a one-factor structure, five-factor structure, and six-factor structure in addition to three bifactor structures. Model-data fit was evaluated based on Comparative Fit Index (CFI), Tucker–Lewis Index (TLI), root mean square error of approximation (RMSEA), and standardized root mean square residual (SRMR). Acceptable fit can be achieved by CFI and TLI values above 0.90, along with RMSEA and SRMR values less than 0.08 [34]. Multigroup CFA was used to evaluate the invariance of the best fitting structure of the IES-R across groups of gender, age, and marital status (see Table 1 for categories). Four models were tested to examine whether the same number of factors are expressed in groups (configural invariance), items load equally on corresponding factors across groups (metric invariance), true mean differences do not vary as a result of constraining intercepts of the regression equations of items on the corresponding factors to equality across groups (scalar invariance), and items display uniqueness (strict invariance) [35,36]. Sample size may affect the values of chi square (χ^2^) index, resulting in the disqualification of fitting models, which demonstrate minor modifications. Therefore, non-invariance was evaluated based on absolute fit indices, which are less likely to be affected by sample size. Particularly, models were deemed non-invariant based on a significant χ^2^, along with ΔCFI and ΔRMSEA above 0.02 and 0.015, respectively. In CFA and multigroup CFA, suggestions indicated by modification indices were performed in order to improve model fit.

Coefficient alpha, item-total correlations, and alpha if item deleted were used to assess the internal consistency of the IES-R. Spearman′s r correlation between the IES-R and its subscales was conducted to determine convergent validity. Spearman′s r correlation between the IES-R and criterion variables (the DASS-8 and its subscales, perceived vulnerability to COVID-19, and perceived physical health status) was used to reflect on the criterion validity of the IES-R. Discriminant validity of the IES-R was evaluated in sample 1 and sample 2 by the heterotrait–monotrait (HTMT) ratio of implied correlations associated with CFA [28]. Known-group validity of the IES-R was evaluated by Mann–Whitney U test, which was employed to compare IES-R scores between quarantined and non-quarantined participants as well as between non-quarantined healthy individuals and psychiatric patients [29,36]. A path analysis structural equation model (SEM) was conducted to examine the relationship among factors comprising the IES-R in the three samples. Analyses were conducted in SPSS and AMOS version 24, and significance was considered at 0.05 two-tailed.

## 3. Results

### 3.1. Characteristics of the Participants

As shown in Table 1, the quarantine sample comprised more males than females. Meanwhile, the majority of the participants in sample 1 and sample 2 were females. The age of around half the participants ranged between 18 and 30 years. Most participants in all the samples had a bachelor’s degree. In the quarantine sample, 7.9% (*n* = 17) of the respondents reported a pre-existing psychiatric disorder. As for sample 1, depression, generalized anxiety disorder (GAD), and obsessive–compulsive disorder (OCD) prevailed in 54.9%, 50%, and 20.5% of the participants, respectively. Few patients were diagnosed with bipolar disorder (6.6%), personality disorders (7.4%), eating disorders (5.7%), sleep disorders (4.1%), and psychotic disorders (2.5%). A considerable number of patients had dual diagnosis, e.g., GAD and/or sleep disorders on top of depression or OCD. More participant characteristics are reported elsewhere [6,29].

### 3.2. Factor Structure of the Arabic Version of the Impact of Event Scale-Revised (IES-R)

#### 3.2.1. Exploratory Factor Analysis

EFA using maximum likelihood with varimax rotation revealed that the items of the IES-R in the quarantine sample covered five factors with eigenvalues > 1, accounting for 62.0% of the variance. Table 2 shows that several items cross-loaded on two or more factors. The KMO test indicated that the participant-to-item ratio was proper for an EFA test. Likewise, Bartlett′s test denoted appropriateness of the sample size for EFA analysis (Table 2).

#### 3.2.2. Confirmatory Factor Analysis

Several competing models of the structure of the IES-R were tested in this study. Appendix A describes the factor structure and items on each factor in every tested model. Table 3 shows that the one-factor structure of the IES-R (Model 1) expressed poor fit in both samples in terms of low CFI and TLI and high RMSEA and SRMR, along with several correlated errors. Model 2 was used to evaluate the proposed three-factor structure of the IES-R (intrusion, avoidance, and hyperarousal). This model and the bifactor structure based on this model had unsatisfactory fit, particularly in the clinical sample. Guided by the results of EFA (see the footnote of Table 2), we tested a five-factor structure (Model 4) including avoidance, intrusion, hyperarousal, difficulty sleeping and concentrating, and numbing. As displayed in Table 3, this model expressed a poor fit. Modification indices revealed that several items had high cross-loadings. Therefore, we kept the five-factor structure but modified the model to allow items with cross-loadings to load only on factors to which they expressed the highest loadings (Model 5). This model has been further modified by forcing items with lower loadings on their domain-specific factors to load on more relevant factors—sleep and irritability/dysphoria appeared as distinct factors in Model 7.

The bifactor structures examined in Model 6 and Model 8 (five and six domain-specific factors along with a general factor) expressed the best fit in sample 2, with all the items of the IES-R loading significantly on the general factor and their domain-specific factors. However, in both models, the loadings of item 20 were less than 0.2 and the loadings of items 7 and 19 were less than 0.3 (Appendix A), indicating weak association with their domain-specific factors. Although Model 6 expressed the best acceptable fit in the clinical sample, item 21 failed to load on the numbing factor, and all the items of the intrusion factor failed to load on their specific factor. Standard errors associated with the items of the intrusion factor were considerably greater than 1 (16.1 to 95.3). In addition, SRMR was not produced due to failure of this model to properly converge. Likewise, Model 8 expressed the best fit compared with other models tested in sample 1. However, item 20 did not load significantly on its corresponding intrusion factor while the loadings of items 3, 19, and 21 on their domain-specific factors were less than 0.3 (Appendix A), denoting that these items had limited contributions to the variance explained by the corresponding factors. The second-order factor structure (Model 9) had a fit slightly lower than that of the six-factor structure in both samples. Accordingly, the six-factor structure seems to represent the best fit in both samples (Figure 1).

#### 3.2.3. Invariance of the Six-Factor Structure of the Impact of Event Scale-Revised (IES-R) across Different Groups

Based on model fit indices, Model 7 was examined for measurement invariance across groups of gender, age, and marital status in both samples. Multigroup CFA showed that the six-factor structure of the IES-R holds configural, metric, and scalar invariance across different groups in both samples, i.e., this model is generally fitting similarly across men and women, young and older, as well as single and married participants. There was a tendency toward strict non-invariance across groups of age and marital status in both samples and across gender in sample 2—as noted by ΔCFI greater than 0.02. However, ΔRMSEA was within the acceptable range (Table 4 and Table 5).

Scalar invariance expressed by this model indicates the suitability of the IES-R to portray variations in true mean differences between groups. Accordingly, Mann–Whitney U test was used to compare differences in the IES-R and the underlying six latent constructs between groups. In sample 1, there were no significant differences in the scores of the IES-R and most of its subscales between men and women, single and married, or individuals aged 30 years or below and those older than 30 years. Only women and participants aged 30 years or below expressed significantly higher levels of irritability/dysphoria than men and older participants (U = 2233.0, 2744.5; z = −2.4, −2.5; *p* = 0.015, 0.012, respectively). As for sample 2, the scores of the IES-R and its subscales, except for sleep disturbance, were significantly higher among women than men (all *p* values = 0.001). The scores of the IES-R, intrusion, hyperarousal, and irritability/dysphoria were significantly higher among single and younger participants in this sample (all *p* values < 0.01). Single participants also expressed significantly higher levels of sleep disturbance (*p* = 0.002). Values of the Mann–Whitney U test and related z scores are shown in Appendix A.

### 3.3. Reliability, Convergent Validity, Normality, Criterion Validity, and Known-Group Validity of the Arabic Version of the Impact of Event Scale-Revised (IES-R)

In order to identify items that may represent a source of misfit, we checked scale reliability and the range of alpha if item deleted. Reliability analysis revealed excellent internal consistency of the IES-R in the quarantine sample, sample 1, and sample 2 (α = 0.90, 0.93, and 0.92, respectively) while alpha if item deleted indicated no further increase in the reliability of the scale following any item removal (range = 0.892 to 0.902, 0.917 to 0.923, and 0.909 to 0.915 in both samples, respectively). Table 6 shows good internal consistency of the factors comprising Model 7, albeit the reliability of the numbing factor was fair in the quarantine sample and acceptable in the other samples.

All the six factors had strong positive correlations with the total scores of the IES-R in the three samples (r ranges between 0.653 and 0.867). Shapiro–Wilk W test shows that the IES-R and its underlying six factors displayed a non-normal distribution, with all *p* values < 0.001. As hypothesized, strong positive correlations were expressed among the scores of the IES-R and the scores of the DASS-8 and its subscales in all the samples (r ranges between 0.176 and 0.716, all *p* values < 0.01). Perceived health status was negatively correlated with the IES-R and its subscales. Meanwhile, the IES-R scores were positively correlated with perceived vulnerability to COVID-19 (Table 6). These results reflect sound criterion validity of the IES-R in all the samples.

As shown in Table 7, Mann–Whitney U test revealed significantly higher levels of the IES-R, avoidance, hyperarousal, sleep disturbance, and irritability/dysphoria among quarantined than community-dwelling participants (*p* values < 0.05). The scores of the IES-R and all its six factors were significantly higher among psychiatric patients compared with those without a psychiatric diagnosis (all *p* values < 0.01), which supports known-group validity of the scale.

The six subscales comprising the Arabic IES-R are distinct from each other in the samples of psychiatric patient and healthy individual, as indicated by all values of the HTMT ratios below 0.85. Only the HTMT ratio between the hyperarousal and intrusion subscales was 0.87 in the community sample, suggesting a trivial overlap—it was less than 0.90 (Appendix A).

### 3.4. Path Analysis Involving the Core Factors Comprising the Arabic Version of the Impact of Event Scale-Revised (IES-R)

In the tested models, all non-significant paths were trimmed. As shown in Appendix A, all path analysis models displayed excellent fit on all indices. In the quarantine sample, the path model explained 38%, 44%, 24%, and 34% of the variances in irritability, sleep disturbance, numbing, and avoidance. Intrusion only had significant direct effects on sleep disturbance and numbing (*p* = 0.048, 0.043); its indirect effects on avoidance and sleep disturbance were non-significant (*p* values > 0.05). The direct effects expressed by hyperarousal and numbing were significant at the level of 0.001 (Figure 2a). Numbing mediated the indirect effects of hyperarousal on avoidance and sleep disturbance (β = 0.109, 0.056; 95% CI: 0.60–0.172, 0.018–0.103; *p* = 0.001, 0.003).

In the psychiatric patient sample, the path model explained 40%, 47%, 29%, and 49% of the variances in irritability, sleep disturbance, numbing, and avoidance. All direct effects were significant at the level of 0.001 (Figure 2b). Irritability mediated the indirect effects of intrusion and hyperarousal on sleep disturbance (β = 0.093, 0.120; 95% CI: 0.046–0.160, 0.056–0.214; *p* values = 0.001). Numbing mediated the indirect effects of intrusion and hyperarousal on avoidance (β = 0.084, 0.067; 95% CI: 0.30–0.173, 0.025–0.132; *p* = 0.004, 0.010).

In the healthy community sample, the path model explained 38%, 42%, 34%, and 45% of the variances in irritability, sleep disturbance, numbing, and avoidance. All direct effects were significant at the level of 0.001 (Figure 2c). Numbing mediated the indirect effects of intrusion and hyperarousal on avoidance (β = 0.163, 0.101; 95% CI: 0.132–0.197, 0.069–0.132; *p* values = 0.001) and on irritability (β = 0.065, 0.040; 95% CI: 0.041–0.094, 0.025–0.061; *p* values = 0.001). Irritability mediated the indirect effects of intrusion, hyperarousal, and numbing on sleep disturbance (β = 0.041, 0.064, 0.027; 95% CI: 0.024–0.064, 0.038–0.097, 0.014–0.046; *p* values = 0.001).

## 4. Discussion

The Arabic version of the IES-R has been previously evaluated only through EFA among 40 Arab refugees in Australia. The current study examined the factor structure of the Arabic IES-R (using EFA and CFA), along with its measurement invariance, convergent validity, normality, criterion validity, known-group validity, and discriminant validity. Therefore, the findings complement existing knowledge by reporting on various psychometric properties of the IES-R in three samples (psychiatric patients, healthy adults, and quarantined individuals) from an Arab country within the context of a collective traumatic event (the COVID-19 outbreak and associated lockdown).

This study could not replicate the theory-based three-factor structure of the IES-R, at least among psychiatric patients. As shown in Table 2, most items had dual or triple loadings greater than 0.3 on more than one factor indicating absence of a well-defined structure of the scale. Several competing models tested in CFA revealed an unsatisfactory fit in the clinical sample. The three-factor structure of the IES-R and a bifactor structure based on that model expressed an acceptable fit in sample 2. However, the best fit in both sample 1 and sample 2 was expressed by a six-factor structure (Model 7) comprising avoidance, intrusion, hyperarousal, numbing, sleep disturbance, and irritability/dysphoria. The six-factor structure is superior to other models tested in this study because items have moderate to strong loadings on their domain-specific factors (Figure 1), and they do not cross-load, signifying fulfillment of the assumption of local independence. Despite the superior fit of the bifactor structure of this model in both samples, some items had low loadings on their domain-specific factors, discouraging the use of the bifactor model in subsequent analysis. Our findings are consistent with many previous studies, which reported a deviation of the dimensionality of the IES-R from the theory-based three-factor structure [16,20,23,25,26].

The literature shows a considerable debate on the dimensional structure of the IES-R. Until the current moment, there has been less consensus among trauma scientists on the best structure of symptom clusters involved in the diagnosis of PTSD [16,20,25,26]. In this respect, investigations of PTSD criteria, as defined in the Diagnostic and Statistical Manual of Mental Disorders (DSM-IV; American Psychiatric Association, 1994), among 1218 women exposed to varying levels of sexual harassment at the workplace denote failure of the intrusion, avoidance, and hyperarousal structure to depict the full picture of the disorder among these women. Alternatively, the authors suggested that PTSD is suitably described within the context of four factors: re-experiencing, effortful avoidance, emotional numbing, and hyperarousal [37]. In the same way, Baschnagel and colleagues evaluated PTSD among western New York undergraduate students one and three months after the 11 September 2001 terrorist attacks. They reported poor fit of the intrusion, avoidance/numbing, and hyperarousal structure of PTSD. Instead, they reported over-time stability of a four-factor model comprising intrusion, avoidance, dysphoria, and hyperarousal. Dysphoria comprised symptoms from hyperarousal and numbing clusters [38]. Subsequent investigations failed to produce a good fit for four-factor structures of the IES-R, which comprised standalone factors of numbing [23,25] or dysphoria [25]. However, best fits were obtained from five-factor models including intrusion, avoidance, hyperarousal, and sleep problems in addition to numbing [25,26] or dysphoria [25]. On the other hand, sleep-related symptoms emerged as an independent factor in investigations involving survivors of a destructive earthquake [20], fire fighters and students [23], as well as survivors of war [26]. A previous preliminary evaluation of the Arabic version of the IES-R also included sleep as a separate construct [24]. However, the sleep construct in our samples comprised only item 2 and item 15. On the other hand, the loadings of item 20 “had dreams about it” on the sleep factor were considerably low while the loadings of this item on the intrusion factor were above 0.4 in both samples. Having a closer look at the content of this item, dreams may be considered a form of unconscious intrusion. Indeed, the literature portrays strong links between cognitive symptoms, sleep problems, and mood dysregulation [13,39].

In our study, irritability, numbness, and sleep were distinct factors, albeit associated with one another. Although irritability and sleep comprised only two items, their internal consistency, item loadings, and item-total correlations were all high (Table 6, Figure 1), supporting these experiences as vivid aspects of PTSD. Collective research suggests that the DSM-IV does not address all the psychological symptoms of PTSD [40]. Meanwhile, a huge number of models evaluating DSM-IV criteria of PTSD report numerous factors, with almost all factors comprising only two symptoms, and high inter-factor correlations are the norm [41]. The distinction between irritability and numbness can be understood within the different contexts of trauma-related emotional responses. General emotional reactions associated with traumatic events mostly comprise anger, fear, sadness, and shame. However, some individuals may be unable to identify their feelings, or they associate strong feelings with past trauma and get overwhelmed with thoughts that expressing negative feelings is dangerous or may create more severe emotions (e.g., a sense of “being out of control/losing it/or going crazy”). Others may express numbness or lack of emotions as they unconsciously deny their trauma-related feelings [40].

Noteworthily, the fit of all tested models was improved by correlating several error terms. However, the fit of the structure expressed by Model 7 (the six-factor structure) in the absence of error correlations was the closest to acceptable fit in sample 1 (χ^2^ (194) = 376.248, CFI = 0.891, TLI = 0.870, RMSEA = 0.075, SRMR = 0.0630) and sample 2 (χ^2^ (194) = 1070.016, CFI = 0.904, TLI = 0.886, RMSEA = 0.068, SRMR = 0.0631). In the meantime, correlated error terms in Model 7 were minimal, denoting a lower chance for items from different factors to flag something other than their respective latent variables. In sample 1, the fit of Model 7 was improved by correlating the error term of item 5 “avoided letting myself get upset” with those of items 4 “was irritable or angry” and 6 “thought of it when I didn′t mean to”. Item 16 “waves of strong feelings” was also correlated with the error of item 19 “reminders of it caused physical reactions e.g., sweating, dyspnea”. In sample 2, the error term of item 5 was correlated with the errors of item 1 “reminder brought back feelings”, item 3 “making me think about it”, items 4, 6, and 9 “images popped into my mind”. Multicollinearity generally suggests the presence of extra latent structures. However, the interactions taking place among these variables may be triggered by method effects, i.e., the sequential appearance of these symptoms on the questionnaire. Alternatively, the wording of these variables denotes that cognitive activation may be evoked by intrusive thoughts about COVID-19 (items 6 and 19). Such cognitive activation may stimulate uncomfortable emotional (items 4 and 16) and physical reactions (item 19). Accordingly, individuals employ avoidance to escape negative emotions and uncomfortable sensory symptoms associated with intrusive thoughts (item 5) or reminders (item 19). Path analysis investigating the relationships among the six factors comprising the Arabic version of the IES-R conforms with this logic. In consistence with an existing Italian study [7], intrusion was associated with hyperarousal in all the samples (Figure 2), and both factors directly and indirectly predicted avoidance, irritability, and sleep disturbance. Their indirect effects were mediated by irritability and numbing. The effects of intrusion on other dimensions of the IES-R were most pronounced in the healthy and clinical samples (who were community residents) than in the quarantine sample (Table 7 and Figure 2). This result draws support from previous investigations showing protective effects of being in isolation against COVID-19-related trauma and distress [6].

The literature refers to the COVID-19 pandemic as a collective trauma, with people experiencing traumatizing information (e.g., about job loss, food shortage, rising prices, etc.) and pictures/videos of health professionals in their quarantine uniform dragging COVID-19 victims on trollies (e.g., on the news and social media) on a daily basis [4]. Although some individuals may be overwhelmed by the trauma, others may get desensitized over time [5]. The six-factor structure was obtained in psychiatric patients and healthy adults. This finding is consistent with those of Gargurevich and colleagues indicating no difference in the overall structure of the IES-R between fire survivors and university students [23]. As shown in Figure 1, most item loadings were a little bit greater in the clinical sample than in the general public sample. The scores of all the six factors were also greater in the clinical sample (Table 7). These results denote that innate stress endorsed by psychiatric patients may increase the intensity of their experience of PTSD [20].

Analysis of measurement invariance revealed that the six-factor structure of the IES-R was invariant at the configural, metric, and scalar levels across groups of gender, age, and marital status. This result is similar to that reported by Wang et al. [20] who found that a four-factor structure model holds invariance across men and women. However, in their study, item intercepts based on magnitudes of factor loadings across samples were significantly non-equivalent, denoting a degree of scalar non-invariance [20]. Scalar invariance of the IES-R expressed in the present study allowed us to compare IES levels between groups. Women and young participants expressed considerably higher levels of trauma than men and older participants in all the samples (Table 7). In accordance, comparisons involving the parents of children diagnosed with epilepsy reflect a higher prevalence of PTSD among women. Men and women experiencing PTSD are reported to exhibit obvious differences in symptom clusters except for avoidance, with women being high on cognitive and somatic symptoms [42]. Irritability, mood dysregulation, and sleep disturbance are associated with continuous activation of the sympathetic nervous system and persistent dysregulation of the hypothalamic–pituitary–adrenal axis [43]. Research reports varying levels of corticotropin-releasing factor level and their association with PTSD symptoms across men and women [18]. Therefore, the differences between men and women in our samples as well as in other studies reflect higher susceptibility of females to react more excessively to traumatic events than their male counterparts.

All the items of the IES-R had strong significant positive correlations with the total scores of the IES-R and their domain-specific factors. Values of alpha if item deleted indicated no further increase in scale reliability following any item deletion. Item-total correlations as well as correlations of the factors with the total score of the scale were high, denoting adequate convergent validity of the IES-R, with no need to remove any item from the scale/subscales. External validity of the IES-R was supported by its positive correlation with the DASS-8, its subscales (depression, anxiety, and stress), as well as perceived vulnerability to COVID-19. Its negative correlation with perceived health status is consistent with the literature reporting higher COVID-19 trauma among people with poor physical health [6]. The IES-R scores were significantly higher among quarantined individuals and psychiatric patients compared with non-quarantined individuals and the general public, which lends further support to its known-group validity. The subscales of the IES-R were not overlapping with one another, as indicated by HTMT ratios below 0.85 in sample 1 and sample 2 (Appendix A). This result supports the discriminant validity of the scale as well as usability of the six subscales to capture the distinct manifestations of trauma.

To our knowledge, this study is the first to investigate the psychometric qualities of the Arabic IES-R in three relatively large samples. Including psychiatric patients and the general public permitted us to evaluate if co-morbidity might affect the structure of the Arabic IES-R. Moreover, sophisticated tests were included such as EFA, CFA, measurement invariance, and HTMT. Thus, our study counteracts obstacle endorsed by former investigations of the Arabic IES-R, such as the use of a small sample size and limited methods of analysis [24]. The findings emphasize that the items of the IES-R cover more than three factors—sleep problems, numbing, and irritability/dysphoria seem to be important aspects of PTSD experiences. Despite these strengths, the results should be interpreted with caution because the study has a number of limitations. Using an online survey as a method to recruit respondents may entail selection bias in terms of including more young people who have some affinity to technology and certain economic background (e.g., they afford a mobile device). Older adults are more vulnerable to severe COVID-19, and their emotional reaction to the pandemic may differ from younger groups. However, we could not test sub-group differences in trauma because those above the age of 50 were less presented (6.1, 9.5, and 14.0% of the quarantine, psychiatric, and healthy samples, respectively (see Supplementary Excel 1 for all age categories)). As the survey was disseminated through social media such as Twitter and WhatsApp, individuals taking part in the psychiatric sample may express lower severity than those who decided not to take part in the survey. The cohorts included were highly heterogenous. For example, females represented a majority in all samples. As noted above, gender may considerably contribute to differences in PTSD symptoms. Data were collected during a restricted period of time at the beginning of the COVID-19 pandemic (29 April–19 May 2020). The number of all confirmed cases in Saudi Arabia did not exceed 1000 at that time. However, the number of cases and disease fatalities have increased given that the pandemic is still ongoing. Therefore, the current levels of psychological distress and trauma associated with the outbreak may be greater than those reported in the present study. In addition, the IES-R was evaluated within the COVID-19 context and extending its functionality to other traumas such as violence, burn, and route traffic accidents may be necessary. More robust methods such as item response theory may be used in future studies to explore the definite relationships among the items of the IES-R and the latent constructs (e.g., item difficulty and discrimination).

## 5. Conclusions

The Arabic IES-R is best described by a six-factor structure comprising avoidance, intrusion, numbing, hyperarousal, sleep problems, and irritability/dysphoria. The scales are quite discrete, and their scores are higher among psychiatric patients, supporting the known-group validity of the IES-R. The scale expresses excellent internal consistency and strong convergent and criterion validity both in psychiatric patients and in the general public. It operates evenly across groups of gender, age, and marital status, which indicates its appropriateness as a measure of psychological trauma among different groups.

## Figures and Tables

**Figure 1 jpm-12-00681-f001:**
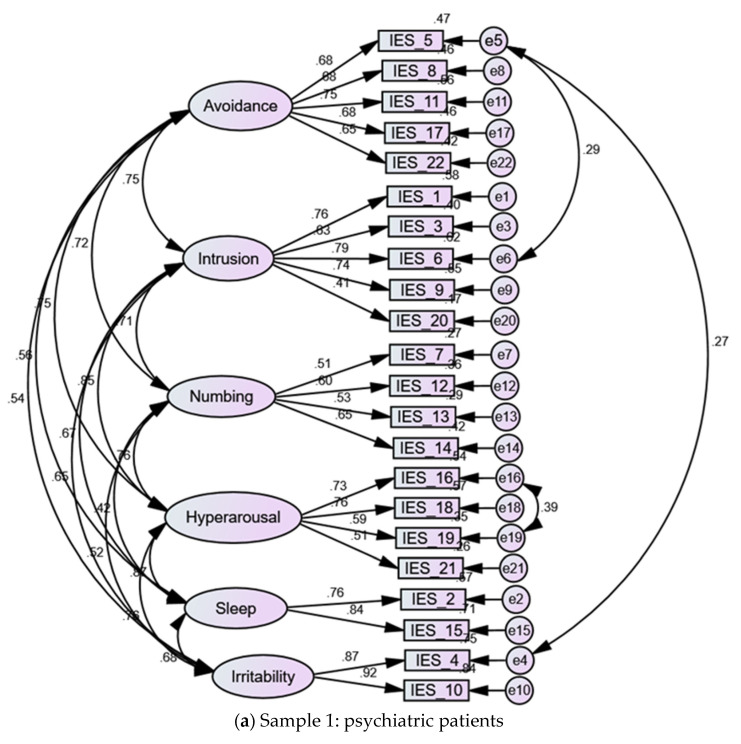
Six-factor structure of the Arabic version of the Impact of Event Scale-Revised (IES-R) among psychiatric patients (**a**) and the general public (**b**). All items had moderate to strong loadings (>0.3) on the corresponding factors as indicated by values on the arrows connecting the factors to the items of the IES-R. The values on the arrows associating the factors of the IES-R reflect considerable inter-factor correlations. The fit of this model was improved by correlating the error term of item 5 with other items in both samples as well as the error terms of items 16 and 19 in the psychiatric patient sample.

**Figure 2 jpm-12-00681-f002:**
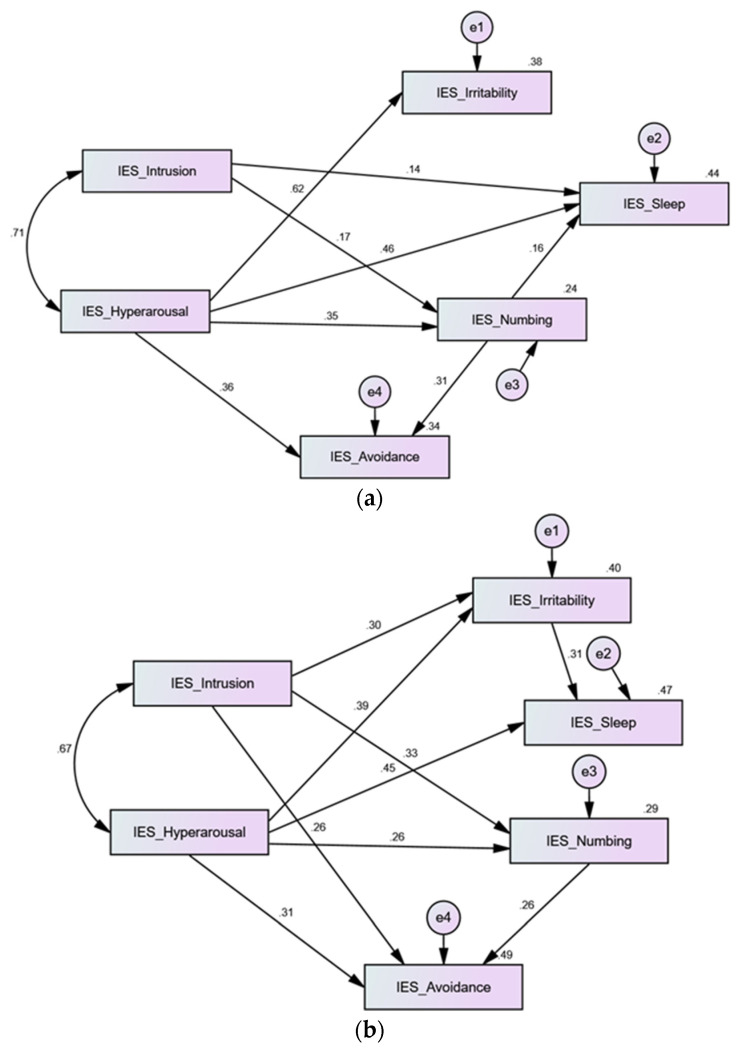
Path analysis model predicting the relationships among factors comprising the Arabic version of the Impact of Event Scale-Revised (IES-R) in the quarantine sample (**a**), psychiatric patients (**b**), and healthy individuals (**c**). The intrusion and hyperarousal subscales of the IES-R were used as predictors and other subscales were dependent variables. Direct effects of the predictors are noted by values on the arrows while indirect relationships are reported in the text. Most variables predicted avoidance both directly and indirectly.

**Table 1 jpm-12-00681-t001:** Sociodemographic characteristics of included participants.

	Quarantine Sample (N = 214) No (%)	Sample 1 (N = 168) No (%)	Sample 2 (N = 992) No (%)
Gender			
Females	86 (40.2)	119 (70.8)	622 (62.7)
Males	128 (59.8)	49 (29.2)	370 (37.3)
Age (years)			
18–30	120 (56.1)	87 (51.8)	448 (45.2)
>31	94 (43.9)	81 (48.2)	544 (54.8)
Marital status			
Married	106 (49.5)	77 (45.8)	553 (55.7)
Single/widowed/divorced	108 (50.5)	91 (54.2)	439 (44.3)
Education			
School degree	65 (30.4)	51 (30.4)	263 (26.5)
University degree	88 (41.1)	105 (62.5)	605 (61.0)
Post-graduate degree	61 (28.5)	12 (7.1)	124 (12.5)
DASS-8 MD (IQR)	2.0 (0.0–7.0)	9 (2.0–17.0)	2 (0.0–7.0)
Depression MD (IQR)	1.0 (0.0–3.0)	4.0 (1.0–7.0)	1.0 (0.0–3.0)
Anxiety MD (IQR)	0 (0.0–2.0)	3.0 (0.0–6.0)	0 (0.0–2.0)
Stress MD (IQR)	0 (0.0–2.0)	2.0 (0.0–4.0)	0 (0.0–2.0)

MD: median; IQR: interquartile range; DASS-8: Depression Anxiety Stress Scale-8.

**Table 2 jpm-12-00681-t002:** Exploratory factor analysis of the Arabic version of the Impact of Event Scale-Revised (IES-R) in the quarantine sample.

Items			Extracted Factors			
		Factor 1	Factor 2	Factor 3	Factor 4	Factor 5
1	Any reminder brought back feelings about it	**0.667**	−0.009	**0.360**	−0.044	0.187
2	I had trouble staying asleep	**0.303**	0.111	**0.640**	0.132	0.134
3	Other things kept making me think about it	**0.717**	0.154	0.290	−0.030	0.125
4	I felt irritable and angry	**0.310**	**0.776**	0.222	0.175	−0.030
5	I avoided letting myself get upset when I thought about it or was reminded of it	0.259	0.298	0.176	**0.341**	**0.319**
6	I thought about it when I did not mean	**0.505**	0.193	0.046	0.298	0.039
7	I felt as if it hadn′t happened or wasn′t real	0.029	0.082	0.082	**0.381**	0.044
8	I stayed away from reminders of it	0.055	−0.016	0.102	0.086	**0.899**
9	Pictures about it popped into my mind	**0.582**	0.206	0.111	0.203	0.151
10	I was jumpy and easily startled	0.259	**0.938**	0.202	0.079	0.062
11	I tried not to think about it	0.031	0.025	0.077	**0.438**	**0.540**
12	I was aware that I still had a lot of feelings	0.156	0.059	0.128	**0.566**	0.206
13	My feelings about it were kind of numb	0.073	0.057	0.128	**0.395**	0.050
14	I found myself acting or feeling like I was back at that time	**0.396**	0.238	0.184	**0.301**	0.115
15	I had trouble falling asleep	0.249	0.163	**0.787**	0.269	0.081
16	I had waves of strong feelings about it	**0.730**	0.260	0.275	0.284	0.086
17	I tried to remove it from my memory	0.297	0.062	−0.107	**0.562**	0.344
18	I had trouble concentrating	**0.348**	0.258	**0.631**	0.136	0.063
19	Reminders of it caused me to have physical reactions	**0.569**	0.158	**0.319**	0.140	0.109
20	I had dreams about it	**0.374**	0.254	0.050	0.175	-0.020
21	I felt watchful and on-guard	0.227	**0.337**	0.045	**0.316**	0.259
22	I tried not to talk about it	0.268	0.083	0.106	0.266	**0.537**
	Kaiser–Meyer–Olkin test	0.870				
	Bartlett′s Test of Sphericity	2219.657				
	Df	231				
	P	<0.001				

Values in boldface represent significant loadings (>0.3). Based on the highest loadings of items with cross-loadings, items were located for the following factors: factor 1, intrusion: items 1, 3, 6, 9, 14, 16, 19, 20; factor 2, hyperarousal/dysphoria: items 4, 10, 21; factor 3, difficulty sleeping/concentrating: items 2, 15, 18; factor 4, numbing: items 7, 12, 13, 17; factor 5, avoidance: items 5, 8, 11, 22.

**Table 3 jpm-12-00681-t003:** Goodness-of-fit indices for models of the Arabic version of the Impact of Event Scale-Revised (IES-R) tested by confirmatory factor analysis in psychiatric patients and healthy adults.

Models	Sample	χ^2^	*df*	CFI	TLI	RMSEA	RMSEA 90% CI	SRMR
Model 1 1F	Sample 1	464.786	205	0.844	0.824	0.087	0.077 to 0.098	0.0714
	Sample 2	1540.825	205	0.853	0.835	0.081	0.077 to 0.085	0.0654
Model 2 3F	Sample 1	409.388	203	0.876	0.859	0.078	0.067 to 0.089	0.0679
	Sample 2	1005.304	202	0.912	0.899	0.063	0.059 to 0.067	0.0596
Model 3 3F bifactor	Sample 1	404.426	202	0.879	0.861	0.077	0.066 to 0.088	0.0679
	Sample 2	1069.021	202	0.905	0.891	0.066	0.062 to 0.070	0.0610
Model 4 5F EFA	Sample 1	393.751	197	0.882	0.862	0.077	0.066 to 0.088	0.0744
	Sample 2	1241.587	197	0.885	0.866	0.073	0.069 to 0.077	0.0751
Model 5 5F modified	Sample 1	382.891	198	0.889	0.871	0.075	0.063 to 0.086	0.0613
	Sample 2	977.984	198	0.914	0.900	0.063	0.059 to 0.067	0.0497
Model 6 5F bifactor	Sample 1	300.239	176	0.925	0.902	0.065	0.052 to 0.077	--
	Sample 2	921.433	197	0.921	0.907	0.061	0.057 to 0.065	0.0528
Model 7 6F	Sample 1	341.248	191	0.910	0.891	0.069	0.057 to 0.080	0.0616
	Sample 2	930.628	189	0.919	0.901	0.063	0.059 to 0.067	0.0573
Model 8 6F bifactor	Sample 1	336.493	190	0.912	0.893	0.068	0.056 to 0.080	0.0765
	Sample 2	895.795	190	0.923	0.906	0.061	0.057 to 0.065	0.0601
Model 9 6F second order	Sample 1	336.394	200	0.899	0.883	0.071	0.060 to 0.082	0.0656
	Sample 2	1063.776	198	0.905	0.889	0.066	0.63 to 0.070	0.0641

*χ*^2^: chi-square; df: degrees of freedom; CFI: Comparative Fit Index; TLI: Tucker–Lewis index; RMSEA: root mean square error of approximation; CI: confidence interval; SRMR: standardized root mean residual; F: factor; --: SRMR was not produced indicating inadequate convergence of that model.

**Table 4 jpm-12-00681-t004:** Invariance of the six-factor structure (Model 7) of the Arabic version of the Impact of Event Scale-Revised (IES-R) across groups of gender, age, and marital status among psychiatric patients.

Groups	Invariance Levels	χ^2^	df	*p*	Δχ^2^	Δdf	*p*(Δχ^2^)	CFI	ΔCFI	TLI	ΔTLI	RMSEA	ΔRMSEA	SRMR
Gender	Configural	699.662	382	0.001				0.831		0.795		0.071		0.0664
	Metric	712.608	398	0.001	12.947	16	0.067	0.832	−0.001	0.806	−0.011	0.069	0.002	0.09701
	Scalar	754.623	419	0.001	42.014	21	0.004	0.821	0.011	0.803	0.003	0.069	0.000	0.0762
	Strict	809.332	444	0.001	54.709	25	0.001	0.805	0.016	0.798	0.005	0.070	−0.001	0.0786
Age	Configural	668.211	382	0.001				0.850		0.819		0.067		0.0867
	Metric	679.811	398	0.001	11.600	16	0.771	0.853	−0.003	0.829	−0.010	0.065	0.001	0.0900
	Scalar	720.006	419	0.001	40.195	21	0.007	0.843	0.010	0.826	0.003	0.066	−0.001	0.1018
	Strict	837.384	444	0.001	117.378	25	0.001	0.794	**0.049**	0.786	**0.040**	0.073	−0.007	0.1134
Marital status	Configural	660.584	382	0.001				0.850		0.819		0.066		0.0847
	Metric	669.533	398	0.001	8.949	16	0.915	0.854	−0.004	0.831	−0.012	0.064	0.002	0.0836
	Scalar	723.437	419	0.001	53.903	21	0.001	0.837	0.017	0.820	0.011	0.066	−0.002	0.0878
	Strict	798.490	444	0.001	75.054	25	0.001	0.810	**0.027**	0.802	0.018	0.069	−0.003	0.0905

*χ* ^2^: chi-square; df: degrees of freedom; CFI: Comparative Fit Index; TLI: Tucker–Lewis index; RMSEA: root mean square error of approximation; CI: confidence interval; SRMR: standardized root mean residual; F: factor. Values in boldface indicate tendency toward non-variance.

**Table 5 jpm-12-00681-t005:** Invariance of the six-factor structure (Model 7) of the Arabic version of the the Impact of Event Scale-Revised (IES-R) across groups of gender, age, and marital status among healthy participants.

Groups	Invariance Levels	χ^2^	df	*p*	Δχ^2^	Δdf	*p*(Δχ^2^)	CFI	ΔCFI	TLI	ΔTLI	RMSEA	ΔRMSEA	SRMR
Gender	Configural	1326.694	378	0.001				0.898		0.875		0.050		0.0585
	Metric	1346.384	394	0.001	19.690	16	0.235	0.897	0.001	0.880	−0.005	0.049	0.001	0.0604
	Scalar	1399.286	415	0.001	52.902	21	0.001	0.894	0.003	0.882	−0.002	0.049	0.000	0.0613
	Strict	1649.733	442	0.001	250.447	27	0.001	0.870	**0.024**	0.864	0.018	0.053	−0.004	0.0613
Age	Configural	1217.433	378	0.001				0.909		0.889		0.047		0.0561
	Metric	1250.212	394	0.001	32.779	16	0.008	0.907	0.002	0.891	−0.002	0.047	0.000	0.0570
	Scalar	1324.160	415	0.001	73.948	21	0.001	0.902	0.005	0.891	0.000	0.047	0.000	0.0649
	Strict	1637.925	442	0.001	313.764	27	0.001	0.871	**0.031**	0.865	**0.026**	0.052	−0.005	0.0593
Marital status	Configural	1242.976	378	0.001				0.906		0.885		0.048		0.0616
	Metric	1287.703	394	0.001	44.727	16	0.001	0.903	0.003	0.886	−0.001	0.048	0.000	0.0602
	Scalar	1354.779	415	0.001	67.076	21	0.001	0.898	0.005	0.887	−0.001	0.048	0.000	0.0678
	Strict	1687.597	442	0.001	332.818	27	0.001	0.865	**0.033**	0.859	**−0.028**	0.053	−0.005	0.0606

*χ* ^2^: chi-square; df: degrees of freedom; CFI: Comparative Fit Index; TLI: Tucker–Lewis index; RMSEA: root mean square error of approximation; CI: confidence interval; SRMR: standardized root mean residual; F: factor. Values in boldface indicate tendency toward non-variance.

**Table 6 jpm-12-00681-t006:** Internal consistency, normality, and criterion validity of the Arabic version of the Impact of Event Scale-Revised (IES-R) and the factors it comprises.

Models	Samples	IES-R	Avoidance	Intrusion	Numbing	Hyperarousal	Sleep Problems	Irritability
Coefficient alpha	Quarantine	0.901	0.778	0.771	0.593	0.729	0.783	0.927
	Sample 1	0.932	0.817	0.808	0.665	0.768	0.773	0.892
	Sample 2	0.915	0.815	0.737	0.681	0.737	0.767	0.872
Alpha if item deleted	Quarantine	0.892–0.902	0.718–0.762	0.681–0.776	0.484–0.565	0.577–0.779	--	--
	Sample 1	0.917–0.923	0.764–0.793	0.737–0823	0.577–0.633	0.661–0.784	--	--
	Sample 2	0.909–0.915	0.738–0.829	0.645–0.750	0.569–0.695	0.633–0723	--	--
Item-total correlations	Quarantine	0.251–0.753	0.496–0.606	0.495–0.668	0.315–0.421	0.353–0.691	0.641	0.865
	Sample 1	0.359–0.722	0.568–0.667	0.389–0.700	0.391–0.477	0.439–0.670	0.636	0.805
	Sample 2	0.316–0.655	0.432–0.650	0.306–0607	0.323–0.531	0.471–0612	0.630	0.773
Correlation with the IES-R	Quarantine	--	0.755 **	0.770 **	0.738 **	0.854 **	0.729 **	0.668 **
	Sample 1	--	0.841 **	0.844 **	0.679 **	0.867 **	0.707 **	0.727 **
	Sample 2	--	0.843 **	0.842 **	0.802 **	0.806 **	0.653 **	0.677 **
Shapiro–Wilk W test	Quarantine	0.971	0.968	0.906	0.926	0.892	0.852	0.753
	Sample 1	0.972	0.964	0.933	0.932	0.927	0.894	0.892
	Sample 2	0.95	0.936	0.899	0.907	0.856	0.752	0.793
Correlation with DASS-8	Quarantine	0.605 **	0.264 **	0.506 **	0.342 **	0.652 **	0.641 **	0.559 **
	Sample 1	0.716 **	0.457 **	0.569 **	0.388 **	0.704 **	0.638 **	0.698 **
	Sample 2	0.623 **	0.408 **	0.523 **	0.417 **	0.614 **	0.508 **	0.647 **
Depression	Quarantine	0.496 **	0.176 **	0.419 **	0.240 **	0.559 **	0.566 **	0.515 **
	Sample 1	0.637 **	0.433 **	0.494 **	0.341 **	0.634 **	0.563 **	0.584 **
	Sample 2	0.555 **	0.359 **	0.469 **	0.380 **	0.547 **	0.452 **	0.565 **
Anxiety	Quarantine	0.551 **	0.290 **	0.515 **	0.267 **	0.569 **	0.450 **	0.564 **
	Sample 1	0.700 **	0.413 **	0.551 **	0.421 **	0.690 **	0.617 **	0.695 **
	Sample 2	0.581 **	0.385 **	0.489 **	0.392 **	0.569 **	0.454 **	0.618 **
Stress	Quarantine	0.574 **	0.253 **	0.438 **	0.396 **	0.603 **	0.598 **	0.497 **
	Sample 1	0.649 **	0.410 **	0.553 **	0.299 **	0.622 **	0.585 **	0.686 **
	Sample 2	0.560 **	0.361 **	0.456 **	0.372 **	0.551 **	0.484 **	0.590 **
Correlation with health status	Quarantine	−0.305 **	−0.038	−0.293 **	−0.227 **	−0.336 **	−0.324 **	−0.389 **
	Sample 1	−0.348 **	−0.204 **	−0.375 **	−0.159 *	−0.303 **	−0.365 **	−0.319 **
	Sample 2	−0.211 **	−0.106 **	−0.199 **	−0.173 **	−0.183 **	−0.189 **	−0.201 **
Correlation with vulnerability	Quarantine	0.136	−0.035	0.238 **	−0.011	0.165 *	0.234 **	0.233 **
	Sample 1	0.236 **	0.104	0.280 **	0.178*	0.223 **	0.208 **	0.142
	Sample 2	0.144 **	0.038	0.182 **	0.062 *	0.186 **	0.179 **	0.172 **

IES-R: Impact of Event Scale-Revised, DASS-8: Depression Anxiety Stress Scale-8 items, MD: median, IQR: interquartile range, * and ** *p* values are significant at 0.05 and 0.01 levels, respectively.

**Table 7 jpm-12-00681-t007:** Descriptive statistics and known-group validity of the Arabic version of the Impact of Event Scale-Revised (IES-R) and the factors it comprises among quarantined, psychiatric, and healthy samples.

IES-R and Its Subscales	Quarantine (N = 214)	Sample 1 (N = 168)	Sample 2 (N = 992)	Quarantined or Not		Having a Psychiatric Disorder or Not	
	MD (IQR)	MD (IQR)	MD (IQR)	Mann–Whitney Test	z	Mann–Whitney Test	z
IES-R	22.0 (12.0–33.0)	30.0 (14.0–43.0)	18.0 (7.0–29.0)	111309.0	−2.4 *	73429.5	−7.3 **
Avoidance	8.0 (4.0–11.0)	8.0 (4.0–12.0)	6.0 (1.0–10.0)	101788.5	−4.2 **	86858.5	−4.6 **
Intrusion	3.0 (1.0–6.0)	5.0 (2.0–9.0)	3.0 (1.0–6.0)	123375.5	−0.1	82829.5	−5.5 **
Numbing	4.0 (1.0–6.0)	4.0 (2.0–7.0)	3.0 (0–6.0)	120033.0	−0.8	90113.5	−4.0 **
Hyperarousal	3.0 (1.0–6.0)	4.0 (2.0–8.0)	2.0 (0–4.0)	112402.5	−2.2 *	74611.0	−7.1 **
Sleep disturbance	1.0 (0–4.0)	2.0 (0–5.0)	0 (0–2.0)	102192.5	−4.4 **	71959.0	−8.0 **
Irritability	0.0 (0–3.0)	3.0 (0–4.0)	1.0 (0–3.0)	114191.5	−2.0 *	78877.0	−6.6 **

IES-R: Impact of Event Scale-Revised, MD: median, IQR: interquartile range, * and ** *p* values are significant at 0.05 and 0.01 levels, respectively.

## Data Availability

The dataset used to produce the current study can be found in Mendeley repository at: https://data.mendeley.com/datasets/8k3vmfxpd3/draft?a=67415321-61f7-4920-bd2a-749b365ff6fb (accessed on 18 March 2022).

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
