# Peer review of "The Arabic Version of the Impact of Event Scale-Revised: Psychometric Evaluation among Psychiatric Patients and the General Public within the Context of COVID-19 Outbreak and Quarantine as Collective Traumatic Events"

_jpm, 2022, doi:10.3390/jpm12050681_

Round 1

Reviewer 1 Report

Please find my comments in the attached file.

Author Response

Journal pf personalized medicine – Manuscript ID: jpm-1668770

Response to the comments of Reviewer 1

We are very much grateful for the time and sincere help of the Reviewer. We have modified the whole manuscript taking into account the comments of the Reviewer, which are addressed line-by-line as shown below. Replies come underneath in red.

Broad Comments

Especially in the context of the COVID-19 pandemic, but certainly not limited to this traumatic event, it is critical to improve measures to identify vulnerable individuals suffering psychological distress following trauma in order to provide the needed support. This study performs a comprehensive analysis and comparison of several models based on the inclusion of different relevant factors and is backed up by thorough statistical testing. The introduction is very informative and provides details about the goals and clearly lays out the rational of the study. The used language is appropriate and only needs some minor revision. However, the study also shows several points of concern and limitations which are outlined below:

We need to note that we had to considerably reduce the text in the Introduction as per a single comment of Reviewer 2.

  • Limitations:

The study has several limitations that are not addressed in the manuscript at all. These include but may not be limited to:

  • The health status of the participants was only assessed by self-evaluation (“rate your physical health status on a scale from 1=very bad to 5=very good). Thus, the gathered data on this topic are subjective and may distort the results.
  • The cohorts are very heterogenous in terms of gender (significantly more females in Sample 1).

1.3 The majority of participants is very young (50% are 18-30 years old) and the age of the rest of the participants is only very vaguely defined (above 31). The latter age range is very wide and given that COVID-19 poses a special risk for the elderly (above 60), it would be important and relevant to subdivide this age group.

1.4 The way of participant recruitment harbors some problems and may also underlie the observed age bias towards young people since invitations to participate were disseminated using social media such as Twitter and WhatsApp. This might also cause a further selection of people with a certain economic background (they can afford a mobile devise) and have some affinity to technology.

1.5 Samples were collected during a very restricted time period at the beginning of the COVID-19 pandemic (April 29 - May 19), thus the time period is very narrow, and it should be mentioned that with the pandemic still ongoing the caused psychological distress might be even larger now.

We acknowledge all the limitations that the reviewer has spotted. The results may be considerably biased by the inclusion of females and youth as well as online survey as a method to recruit respondents during a restricted period of time at the beginning of the COVID-19 pandemic (April 29 - May 19, 2020). We have addressed all these limitations should in the Discussion (Line 584-602).

  • References:

The references contain many self-citations (nearly ¼ of all references) and in some instances, the cited articles seem to be inappropriate, for example ref. 5 line 65, ref. 14 line 77, ref. 15 line 79, ref. 30 line 144, ref. 29, 40 line 493. I would strongly recommend replacing these references by more appropriate articles and limit the number of self-citations.

Yes, we have removed all the references noted in the comment. Other relevant references were included

Specific comments

  • Abstract:
    • Line 41: Depression Anxiety Stress Scale-8

Thank you so much. We have corrected the name of the scale.

1.2 Line 47: Do the authors mean “driving” instead of “deriving”?

Yes, the reviewer is right. The correct word has been used. Thank you so much.

  • Introduction:
    • Line 89: Replace “latter” by “later”
    • Line 92: “5” needs to be replaced by “4”

Thank you so much. The comments are quite relevant. However, the indicated text has been omitted in the revision.

  • Material and Methods:
    • Line 155: Please further define the term “quarantine setting”, it is not clear what it means in this context.

Thank you so much. We have added a brief text describing “quarantine setting” in this context (line 110-115).

  • The number of participants in the quarantine sample should be specified in 1.

Yes, the number of participants in the quarantine sample should be specified in 2.1. (line 108).

  • Line 173-176: sentence needs to be revised

Yes, this sentence has been rewritten. We hope that it is currently better than before (line 133-135).

  • Results

4.1 Line 258: add “group” or “sample” after “quarantine”

Thank you so much, “sample” has been added after “quarantine” (line 219).

4.2 Line 288: remove semicolon (;)

Yes, the semicolon (;) has been removed (line 253).

4.3 Figure 1 and 2: the figures are complex and need further explanation in the figure legends (f. ex. What do arrows and numbers indicate?)

Yes, we have briefly elaborated on the structure of the IES-R presented in the Figure. If the reviewer would request further changes, we would be more than happy to do them.

4.4 Table 5:  

  • title: remove “in” before “among” (line 356)
  • please explain abbreviations and bold numbers in the legends

Thank you so much. “in” has been removed from the title of Table 5.

An explanation of the abbreviations and bold numbers has been added in the legends of Tables 4 and 5. However, we apologize for an extra page was added on typing. We used the document prepared by the copy Editor, and it is not possible to get rid of it with delete or back space. So, they would probably mange this issue during the production phase.

4.5 Please use terms consistently (alpha-if-item deleted [line 377] vs. alpha if item deleted [line 379])

Yes, thank you so much. We have used “alpha if item deleted” consistently in this version.

4.6 Line 395: significant p value p<0.05.

Really grateful for the reviewer’s alert observation. It has been corrected (p values < 0.05).

  • Discussion
    • Line 488: remove comma
    • Line 553: remove “a bit”
    • Line 557: remove “a chance”
    • Line 560: replace “men and women” by “parents”
    • Line 593: replace “emphasis” by “emphasize”

We are quite grateful for helping us remove clutter and correct punctuation/typos. All the words noted have been removed or modified as the reviewer has requested.

  • Abbreviations

US United States (capital letter)

Yes, thank you so much. The name of the country has been corrected.

We hope that the manuscript has been satisfactorily modified and that the current version will be suitable for publication.

Best regards,

Reviewer 2 Report

The logical build-up of the manuscript is coherent and it contains minimal spelling error.

The Introduction is excessively long, probably over a 1300 words. In general, the introduction is normally around 500 words and is a brief introduction to the subject matter and outlining the reason for the study. Please reduce.

Author Response

Journal pf personalized medicine – Manuscript ID: jpm-1668770

Response to the comments of Reviewer 1

The Introduction is excessively long, probably over a 1300 words. In general, the introduction is normally around 500 words and is a brief introduction to the subject matter and outlining the reason for the study. Please reduce.

Thank you so much for such an important comment. We agree with the reviewer that the Introduction was enormous. We have removed unnecessary contents, resulting in a 558-word long Introduction in this revised version. We hope that the manuscript has been satisfactorily modified and that the current version will be suitable for publication.

Best regards,

Round 2

Reviewer 1 Report

Dear Authors, Thank you for carefully addressing all of the reviewers' comments and suggestions. In the revised version, the limitations have been pointed out clearly in the discussion and inappropriate self-citations were removed. All the best for the future.